# Mandarin fish (Sinipercidae) genomes provide insights into innate predatory feeding

Shan He[1,2,6], Ling Li[1,2,3,6], Li-Yuan Lv[1,2,6], Wen-Jing Cai[1,2], Ya-Qi Dou[1,2], Jiao Li[1,2], Shu-Lin Tang[1,2], Xu Chen[1,2], Zhen Zhang[1,2], Jing Xu[1,2], Yan-Peng Zhang[1,2], Zhan Yin[4], Sven Wuertz[3], Ya-Xiong Tao[5], Heiner Kuhl[3✉] & Xu-Fang Liang[1,2✉]

Mandarin fishes (Sinipercidae) are piscivores that feed solely on live fry. Unlike higher vertebrates, teleosts exhibit feeding behavior driven mainly by genetic responses, with no modification by learning from parents. Mandarin fishes could serve as excellent model organisms for studying feeding behavior. We report a long-read, chromosomal-scale genome assembly for *Siniperca chuatsi* and genome assemblies for *Siniperca kneri*, *Siniperca scherzeri* and *Coreoperca whiteheadi*. Positive selection analysis revealed rapid adaptive evolution of genes related to predatory feeding/aggression, growth, pyloric caeca and euryhalinity. Very few gill rakers are observed in mandarin fishes; analogously, we found that zebrafish deficient in *edar* had a gill raker loss phenotype and a more predatory habit, with reduced intake of zooplankton but increased intake of prey fish. Higher expression of *bmp4*, which could inhibit *edar* expression and gill raker development through binding of a Xvent-1 site upstream of *edar*, may cause predatory feeding in *Siniperca*.

[1] College of Fisheries, Chinese Perch Research Center, Huazhong Agricultural University, Wuhan, China. [2] Innovation Base for Chinese Perch Breeding, Key Laboratory of Freshwater Animal Breeding, Ministry of Agriculture, Wuhan, China. [3] Department of Ecophysiology and Aquaculture, Leibniz Institute of Freshwater Ecology and Inland Fisheries, Berlin, Germany. [4] State Key Laboratory of Freshwater Ecology and Biotechnology, Institute of Hydrobiology, Chinese Academy of Sciences, Wuhan, China. [5] Department of Anatomy, Physiology & Pharmacology, College of Veterinary Medicine, Auburn University, Auburn, AL, USA. [6] These authors contributed equally: Shan He, Ling Li and Li-Yuan Lv. ✉email: kuhl@igb-berlin.de; xfliang@mail.hzau.edu.cn

Slow-moving, easily digested and small zooplankton as well as benthic organisms are commonly the first foods consumed by various fishes, such as *Mylopharyngodon piceus*, *Mugil soiuy*, *Macrura reevesii*, and *Hippoglossus hippoglossus*[1–3], in different ecological environments. For example, the first foods of yellow perch (*Perca flavescens*) larvae are copepod nauplii and cyclopoid copepods, and *Bosmina coregoni* becomes the dominant food after the fish reach a length of 11 mm[4]. Similarly, European perch (*Perca fluviatilis*) larvae first feed on *Paramecium caudatum*[5]. Atlantic cod (*Gadus morhua*) larvae feed on cultivated copepods (*Acartia tonsa*) at the first feeding[6]. Although the adults of these piscivorous fishes feed on other fish, their larvae can feed on zooplankton during the first feeding stages. In contrast, mandarin fishes (Sinipercidae) are innate and obligate piscivores; once their fry start feeding, they feed solely on live fry of other fish species and refuse zooplankton or formulated diets[7,8]. The feeding ability of mandarin fish larvae is affected by the availability of piscine prey. Furthermore, cannibalism occurring at the first feeding in mandarin fishes causes the death of both predator and prey, resulting in great economic losses in aquaculture. Currently, little is known about the molecular genetic factors that underlie predatory feeding and species-specific feeding habits.

In higher vertebrates, such as mammals, the responses of primary centers are coordinated by correlation centers, and the cerebrum is the central region for memory. The results of experiences on which intelligence and learning are based are processed in this region[9]. In fish, on the one hand, learning of feeding habits is influenced by individual experience[10]. On the other hand, genetic factors have a decisive role in shaping fish feeding behaviors. This differs from the process in mammals and birds, where learning of feeding habits is determined by the parents. Therefore, in fish, in contrast to mammals, the study of predatory feeding behavior from first feeding could enable us to learn much more about the direct effects of genes and genetic factors on feeding habits at a large scale. Mandarin fishes might serve as excellent model organisms in this context.

The four species in the Sinipercidae family, *Siniperca chuatsi*, *Siniperca kneri*, *Siniperca scherzeri*, and *Coreoperca whiteheadi*, which are widely distributed in China, exhibit significant differences in biology. Compared with *S. chuatsi*, *S. scherzeri* can easily be weaned onto dead prey fish or artificial diets[11]. *S. kneri* shows low growth performance[12], whereas *C. whiteheadi* possesses fewer pyloric caeca[13], which serves as an adaptation to increase the surface area and nutrient uptake capacity of the fish gut. Furthermore, compared with fish from marine habitats, such as *Dicentrarchus labrax* and *Lates calcarifer*, these four sinipercid fish live in freshwater but still possess the ability to adapt to salinity. However, genomic resources have not been established for Sinipercidae, and their species-specific adaptive evolution regarding predatory feeding, growth performance, pyloric cecum development and euryhalinity tolerance are poorly understood.

*S. chuatsi* is an important aquaculture fish and is highly valued due to its excellent flesh quality. With an output value of more than 3 billion U.S. dollars (20 billion Yuan)[14], the economic impact of *S. chuatsi* aquaculture far exceeds that of species that are produced at similar scales, such as gilthead sea bream and European sea bass. A main factor that limits sustainable aquaculture of *S. chuatsi* is its strict piscivory[15].

Here, we report the genome sequences of four members of the Sinipercidae. Single-molecule real-time (SMRT) long-read sequencing enabled a state-of-the-art chromosomal-scale genome assembly of *S. chuatsi*. The genomes of its close relatives *S. kneri*, *S. scherzeri* and *C. whiteheadi* were sequenced by Illumina sequencing and assembled with high quality in a cost-effective manner. In addition, comparative genomic analysis between mandarin fish genomes and genomes available for other ray-finned fishes (Actinopterygii) allowed us to gain insights into the genetic basis of species-specific biology in mandarin fishes. These results will support aquaculture breeding programs and research on mandarin fishes and lay a solid genetic foundation for understanding species-specific adaptations and the evolution of predatory feeding behavior.

## Results

**Genome assembly of *S. chuatsi*.** SMRT sequencing is the method of choice for de novo assembly of vertebrate genomes. In the case of *S. chuatsi*, Pacific Biosciences P6-C4 chemistry and long-read genome assembly[16,17] delivered primary genome assemblies with N50 contig sizes in the range of 2–6 Mbp. By using remapped raw reads and synteny checks for scaffolding of the primary assemblies and subsequent gap closure, we were able to improve the contig N50 to 12.2 Mbp, which is 4–10 times greater than the values in similar teleost projects. Indeed, we were able to construct 96.7% of the assembled contig sequence length into 24 chromosomes using only a linkage map based on restriction site-associated DNA sequencing (RADseq). For an overview of the genome assembly steps and results, see Supplementary Fig. 1 and Supplementary Tables 1 and 2. Of 2166 linkage group markers (with a mapping quality of MQ60), 2162 (99.82%) could be mapped back to the corresponding assembled chromosome, and only 4 markers (0.18%) mapped to different groups in the linkage map and chromosome assembly. Seventy-six scaffolds (1–8 scaffolds/chromosome) were oriented using this linkage map. We annotated 28406 protein-coding genes that coded for proteins longer than 99 amino acids using evidence from RNA sequencing and protein homology. In addition, we annotated 4444 smaller potentially coding regions.

The completeness of the chromosome assembly is also underlined by the presence of $(CCCTAA)_n$ telomeric repeats close to the terminal regions of the assembled chromosomes. The large fraction of bases assembled into chromosomal sequences, small number of gaps in the chromosomes ($n = 304$) and signatures of telomeric repeats at chromosome ends as well as the very large fraction of complete Benchmarking Universal Single-Copy Orthologs (BUSCO)[18] genes (Actinopterygii set, sinChu7_genome: C: 97.1% [S: 94.7%, D: 2.4%], F: 1.4%, M: 1.5%; SC7_annotation: C: 94.5% [S: 91.0%, D: 3.5%], F: 3.7%, M: 1.8%; C = complete, S = single, D = duplicated, F = partial, M = missing) provide high confidence that this teleost genome assembly is nearly complete.

**Genome assemblies of other three species of Sinipercidae.** The costs of assembling high-quality draft genomes from short reads currently depend on the construction and sequencing of several long-distance mate-pair libraries. We circumvented this by using "gel-free" mate-pair libraries, where no size selection of the template DNA was performed. As undefined insert sizes may result in less efficient scaffolding during genome assembly, we mapped these reads to the high-quality *S. chuatsi* reference genome and sorted mapped mate pairs of a particular insert size range into distinct in silico size-selected files. Our short-read assembly pipeline (Supplementary Fig. 2, Supplementary Tables 1 and 2) revealed 3 high-quality draft de novo genome assemblies for *S. kneri*, *S. scherzeri*, and *C. whiteheadi* with ~40× total sequencing coverage for each species (50% 300 bp paired-end fragments/50% gel-free mate pairs). A very large fraction (>97% of bases) of the de novo-assembled scaffolds (N50 length: 1.0–1.4 Mbp) could be anchored to chromosomal groups when comparing synteny/collinearity with the *S. chuatsi* reference genome. The high mate-pair sequencing coverage resulted in a large contig

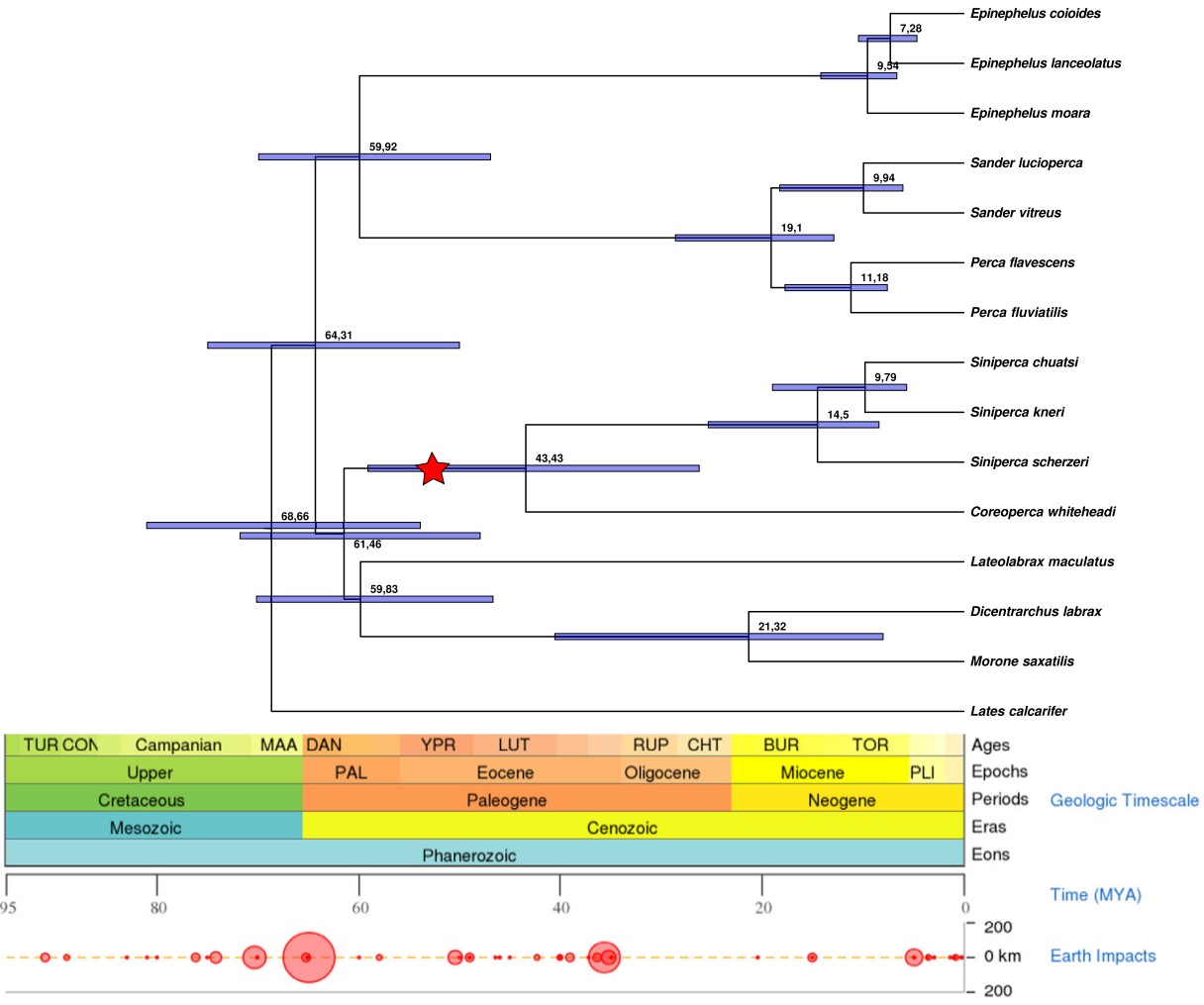

**Fig. 1 Time-calibrated phylogenomic tree calculated from noncoding portions of whole-genome alignments.** The SH-aLRT support was 100 for all branches. Divergence times (red or x axis) were estimated by MCMCTree[62] (clock = 2 model) using a few calibration timepoints from www.timetree.org. The mandarin fish (Sinipercidae) clade is indicated with "★".

N50 length of 76–84 kb (in relation to those for other teleost short-read assemblies).

**Phylogenomic tree.** We produced a multiple whole-genome alignment from 120 publicly available teleost fish genomes (NCBI and Ensembl), added a shark genome (*Callorhinchus milii*) to serve as a highly diverged outgroup, and constructed a phylogenomic tree based on the concatenated noncoding fraction of the alignment (4348668 nt positions) using RAxML[19]. The highly supported resulting tree grouped the Sinipercidae into an independent monophylum (Fig. 1; Supplementary Figs. 3 and 4). The common ancestor of all sinipercid fishes was estimated to have occurred 43.43 million years ago (Mya) (confidence interval (CI): 59.09–26.24 Mya), during the early Eocene. The ancestor of the genus *Siniperca* was estimated to have occurred 14.5 Mya (CI: 25.34–8.44 Mya), during the Neogene. Within the clade, *S. chuatsi* and *S. kneri* diverged from each other 9.79 Mya (CI: 18.97–5.66 Mya).

**Whole-genome alignments.** Alignments between the whole-genome assemblies from the four sinipercid fishes and high-quality assemblies from the more distantly related *D. labrax* and *L. calcarifer* enabled us to visualize principles of chromosomal evolution (Fig. 2a). Here, increasing numbers of chromosomal

rearrangements depending on evolutionary distance from *S. chuatsi* were observed (*S. kneri* = 79; *S. scherzeri* = 76; *C. whiteheadi* = 162; *D. labrax* = 251; *L. calcarifer* = 345). More interestingly, these rearrangements occurred at higher frequencies toward the ends of chromosomes. Chromosome ends also tend to be difficult to align between distantly related species due to high sequence divergence. A possible biological explanation for these findings, in addition to the technical explanation that sub-telomeric regions are highly repetitive and prone to misassembly, is that sequence divergence and rearrangement may be long-term effects of the higher crossing-over frequencies at chromosome ends[20].

**Comparative genomic analyses.** To understand the genetic factors affecting the different evolutionary features among the four Sinipercidae species, we performed an analysis of all annotated protein-coding genes in the *S. chuatsi* reference genome. In this regard, we focused on copy number variation, species-specific genes and positively selected genes (PSGs) to screen for candidate genes for further investigation. In particular, the analysis of episodic diversifying selection by adaptive branch-site random effects likelihood (aBSREL)[21] combined with functional enrichment analysis (Gene Ontology (GO) and mammalian phenotype (MP) databases) provided a global view (Fig. 2b) of the evolution

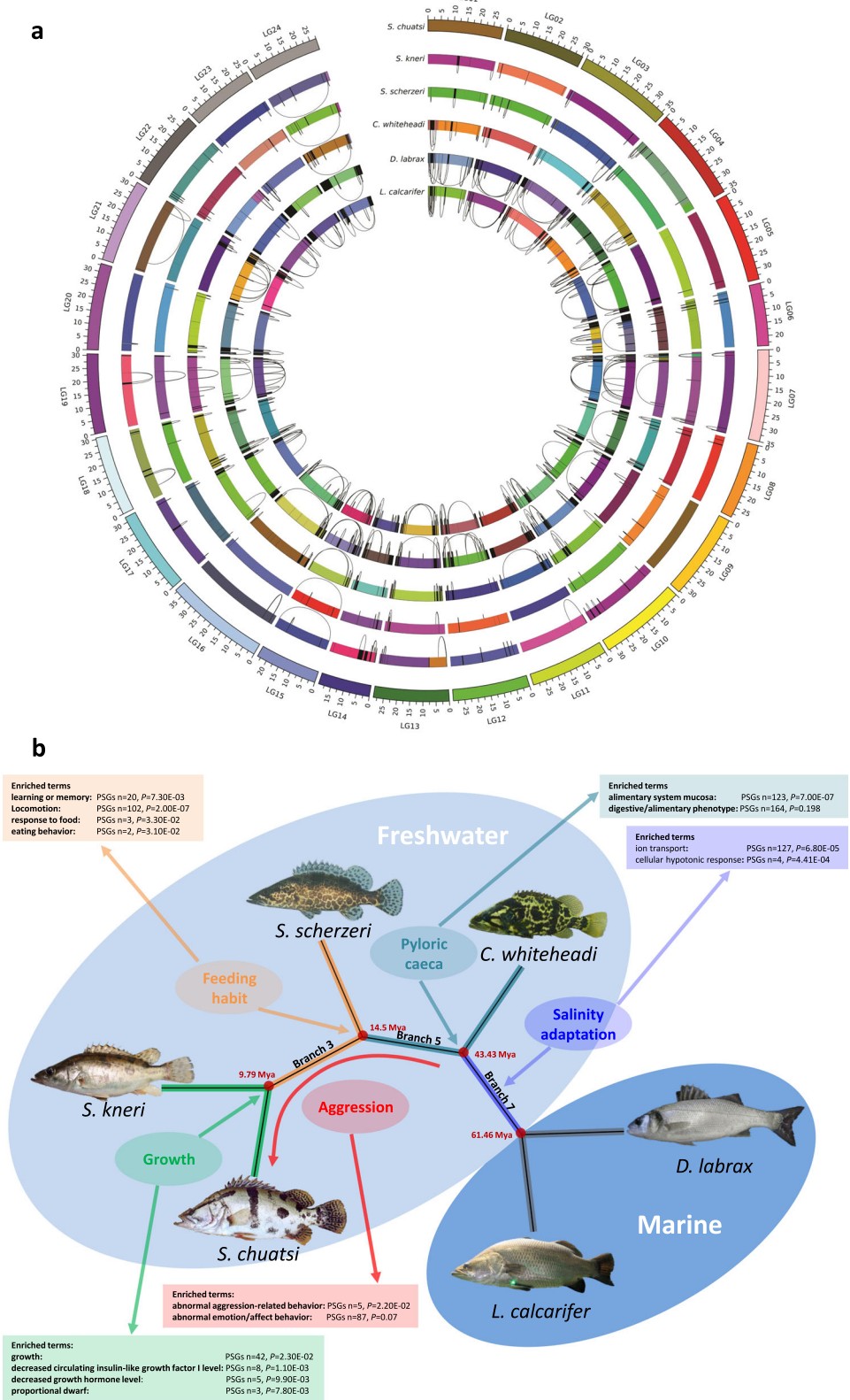

**Fig. 2 Whole-genome alignments. a** Visualization of whole-genome alignments. Chromosomes and scaffolds of each assembly are shown in random colors as syntenic blocks on the reference *S. chuatsi* genome coordinate system. Most chromosomes show highly complete synteny, as depicted by uniform coloring. Intrachromosomal rearrangements between the blocks are connected by black lines. **b** Functional enrichment analysis of genes with signatures of positive selection in the Sinipercidae reveals the episodical adaptive evolution of important traits (for corresponding gene lists, see Supplementary Table 3).

of important traits (feeding habit, growth, pyloric caeca, salinity adaptation, and predatory feeding/aggression behavior) in the three *Siniperca* spp. and *C. whiteheadi* since their split from the two marine outgroup species (*D. labrax* and *L. calcarifer*).

**Feeding habit**. A number of 421 genes on branch 3 and 636 genes on the *S. scherzeri* branch were under positive selection ($P < 0.05$) (Fig. 2b). PSGs with assigned gene symbols ($n = 544$) were subjected to enrichment analysis. The GO term "learning or memory" was enriched for 20 PSGs (uncorr. false discovery rate (FDR) $P = 7.3e-3$, ranked 266th in the list), and the GO term "locomotion" was very highly significant for 102 PSGs (uncorr. FDR $P = 2.0e-7$, ranked 2nd in the list). Considering only PSGs evolving on the branch of *S. scherzeri*, we found that 3 of 334 PSGs were related to "response to food" (uncorr. FDR $P = 3.3e-2$, ranked 583rd in the list). On branch 3, we found that 2 of 84 PSGs (more stringent PSG list $P < 0.005$) were related to "eating behavior" (uncorr. FDR $P = 3.1e-2$, ranked 233rd in the list).

In a previous study, we identified the key genes and pathways, such as learning and memory, involved in the unique food preference through transcriptome analysis of dead prey fish feeders and nonfeeders in hybrid F1 offspring of *S. chuatsi* (♀) × *S. scherzeri* (♂)[11]. Among the PSGs, 12 genes on branch 3 and 20 genes on the *S. scherzeri* branch were differentially expressed between dead prey fish feeders and nonfeeders (log2 fold change (log$_2$FC) ≥ 1) (Supplementary Tables 4 and 5).

**Growth**. To identify the key genes that determine the marked difference of the growth performance between *S. kneri* and *S. chuatsi*, we performed enrichment analysis of 411 PSGs ($P < 0.05$) with assigned gene symbols from the *S. kneri* or *S. chuatsi* branches which revealed 42 genes related to "growth" (uncorr. FDR $P = 2.3e-2$, ranked 177th in the list). More significant results were obtained by querying the MP ontologies, which showed 8 genes related to "decreased circulating insulin-like growth factor I level" (uncorr. FDR $P = 1.1e-3$, ranked 7th in the list) and 5 genes related to "decreased growth hormone level" (uncorr. FDR $P = 9.9e-3$, ranked 84th in the list). Interestingly, considering only PSGs from the *S. kneri* branch, 3 of 206 genes showed enrichment of the MP term "proportional dwarf" (uncorr. FDR $P = 7.8e-3$, ranked 34th in the list).

In a previous study, we identified genes involved in the growth of *S. chuatsi* by transcriptome sequencing[22]. We searched the transcriptome data with these PSGs and identified ten *S. chuatsi*-specific genes that were differentially expressed between groups of fishes with different growth performances (Supplementary Table 6). In addition, 414 genes on the *S. chuatsi* branch and 428 genes on the *S. kneri* branch were under positive selection ($P < 0.05$), of which 7 PSGs in *S. chuatsi* and 18 PSGs in *S. kneri* were also differentially expressed in the growth performance experiments (Supplementary Tables 7 and 8).

**Pyloric caeca**. To identify the functional genes involved in the formation of pyloric caeca, we analyzed positive selection occurring between *Siniperca* sp. and *C. whiteheadi* (*Coreoperca* branch and branch 5). Enrichment analysis of 1687 PSGs revealed 123 genes that were highly significantly associated with the Edinburgh Mouse Atlas Project Anatomy (EMAPA) term "alimentary system mucosa" with high significance (uncorr. FDR $P = 7.0e-7$, ranked 1st in the list). MP term analysis revealed 164 PSGs related to "digestive/alimentary phenotype" at low significance (uncorr. FDR $P = 0.198$, ranked 1856th in the list), 129 of which were also related to the MP term "abnormal digestive system morphology".

One gene cluster consisting of minichromosome maintenance complex component 5 (*mcm5*), 2-amino-3-ketobutyrate coenzyme A ligase, mitochondrial (*gcat*), melanin-concentrating hormone receptor 1-like (*mchr1*), and nucleosome-remodeling factor subunit BPTF-like (*bptf*) was under positive selection in *C. whiteheadi* (Supplementary Table 9). Furthermore, we identified 8 genes involved in the development of the endoderm and the digestive tract that were under positive selection on branch 5 or the *Coreoperca* branch, such as *hoxb9a*, *hoxb2a*, *hoxb1a*, noggin-2-like (*nog2*), insulin gene enhancer protein ISL-1 (*isl1*), *bmp10*, *bmp7b*, and fibroblast growth factor 4 (*fgf4*). In addition, we identified three *pepsin A* genes in *Siniperca* sp. but only two *pepsin A* genes and a pseudogene of *pepsin A* in *C. whiteheadi* (Supplementary Table 10).

**Salinity adaptation**. As the last common ancestor of mandarin fishes and European sea bass was a marine fish, we compared mandarin fishes (freshwater, FW) and European sea bass (seawater, SW) and found that a total of 1382 genes on branch 7 were subjected to positive selection ($P < 0.05$). Of these, 918 genes had assigned gene symbols, and GO term enrichment analysis revealed involvement of 127 genes in "ion transport" (uncorr. FDR $P = 6.8e-5$, ranked 16th in the list). In addition, 4 genes showed enrichment for "cellular hypotonic response" (uncorr. FDR $P = 4.41e-04$, ranked 52nd in the list). The total number of copies of the Na$^+$/K$^+$-ATPase gene (only 12 copies) in mandarin fishes was more similar to that of marine and euryhaline fishes, such as tongue sole (*Cynoglossus semilaevis*, 12 copies), Asian sea bass (*L. calcarifer*, 12 copies), and fugu (*Takifugu rubripes*, 12 copies), than to that of freshwater fishes, such as zebrafish (*Danio rerio*, 16 copies), Mexican cave fish (*Astyanax mexicanus*, 16 copies), and northern pike (*Esox lucius*, 15 copies) (Supplementary Table 11).

*Aqp1a*, *aqp3a*, *aqp3b*, and *aqp8b* were positively selected in *L. calcarifer*, in *D. labrax* and on branch 7. Synteny analysis indicated that *aqp8aa* was absent in the mandarin fishes (freshwater), while *aqp8b* was lost in the cichlid fishes (FW/BW) (Supplementary Table 12 and Supplementary Fig. 5).

**Predatory feeding behavior and predatory aggression**. Gene set enrichment analysis of 212 PSGs on the *S. chuatsi* branch revealed 5 genes related to the MP term "abnormal aggression-related behavior" (uncorr. FDR $P = 2.2e-2$, ranked 164th in the list). One of these genes, *hydin*, seems to be at least partially duplicated in a large tandem duplication (~100 kb) specific to the *S. chuatsi* genome. Enrichment analysis of all PSGs identified in the *Siniperca* clade (e.g., all branches since the split from the last common ancestor of *Siniperca* and *Coreoperca*) ($n = 1528$) revealed 87 PSGs related to the MP term "abnormal emotion/affect behavior" with slightly lower significance (uncorr. FDR $P = 0.07$, ranked 845th in the list). In addition, 4 genes (phenylethanolamine-*N*-methyltransferase (*pnmt*), kin of IRRE-like protein 3 (*kirrel3*), granulin (*grn*), and neuronal PAS domain-containing protein-4 (*npas4*)) with copy number variation between sinipercids and nonpredatory fish species were identified (Table 1). These genes might contribute to the predatory feeding behavior and predatory aggression of mandarin fishes (Fig. 3a).

In a previous study, we designed a specific training paradigm and found that some individuals could accept dead prey fish successfully (feeders), while some could not (nonfeeders)[15]. We searched the PSGs and genes with copy number variation in transcriptome data of feeders and nonfeeders and found that multiple genes, including adenylate cyclase 3 (*adcy3*), Arginine vasotocin/Vasotocin-neurophysin VT 1 (*avt*), estrogen receptor alpha (*esr1*), Follicle-stimulating hormone beta (*fshb*), glutamate

**Table 1 Expansion and loss of predatory feeding genes.**

| Species | Order | pnmt | kirrel3 | grn | npas4 |
|---|---|---|---|---|---|
| Siniperca chuatsi | Centrarchiformes | 3 | 2 | 3 | 1 |
| Siniperca kneri | Centrarchiformes | 2 | 2 | 3 | 1 |
| Siniperca scherzeri | Centrarchiformes | 2 | 2 | 3 | 1 |
| Coreoperca whiteheadi | Centrarchiformes | 2 | 2 | 3 | 1 |
| Dicentrarchus labrax | Eupercaria incertae sedis | 2 | 2 | 3 | 2 |
| Larimichthys crocea | Eupercaria incertae sedis | 2 | 2 | 3 | 2 |
| Lates calcarifer | Carangaria incertae sedis | 1 | 2 | 3 | 2 |
| Nototthenia coriiceps | Perciformes | 2 | 1 | 3 | 2 |
| Stegastes partitus | Ovalentaria incertae sedis | 1 | 1 | 3 | 2 |
| Oreochromis niloticus | Cichliformes | 1 | 1 | 6 | 3 |
| Pundamilia nyererei | Cichliformes | 1 | 1 | 6 | 2 |
| Haplochromis burtoni | Cichliformes | 1 | 1 | 6 | 1 |
| Maylandia zebra | Cichliformes | 1 | 1 | 8 | 3 |
| Neolamprologus brichardi | Cichliformes | 1 | 1 | 4 | 2 |

Fish with predatory feeding and omnivorous feeding are non-underlined and underlined, respectively.

decarboxylase (*gad*), histamine *N*-methyltransferase (*hnmt*), 5-hydroxytryptamine (serotonin) receptor 1B (*htr1b*), *kirrel3*, *npas4*, Isotocin-neurophysin IT 1 (*oxt*), regulator of G-protein signaling 6 (*rgs6*), Tyrosine hydroxylase/Tyrosine 3-monooxygenase (*th*), and amine oxidase [flavin-containing] (*maob*), were differentially expressed between fishes that could and could not feed on dead prey fish or artificial diets (Supplementary Table 13). These 13 genes might be candidate genes controlling the predatory feeding behavior in mandarin fishes. The results of RT-qPCR were consistent with those of transcriptome analysis (Supplementary Fig. 6 and Supplementary Table 14).

**Gill rakers and predatory feeding behavior.** Mandarin fishes feed solely on live fry of other fish and have only 4–9 gill rakers, fewer than other fishes (Supplementary Table 15), reflecting that teleosts with different numbers of gill rakers show different prey spectra and trophic ecologies. To illuminate the mechanism underlying fewer gill rakers in mandarin fishes, we analyzed the *eda* and *edar* gene sequences from mandarin fish genomes and compared their mRNA expression between *S. chuatsi* and zebrafish. The copy number, synteny and genomic structures of *eda* and *edar* were highly conserved (Supplementary Figs. 7 and 8), whereas the absolute mRNA expression levels of the *eda* and *edar* genes in *S. chuatsi* were significantly lower than those in zebrafish (Fig. 3b). We hypothesized that lower *eda* and *edar* expression might contribute to the fewer gill rakers in *S. chuatsi* than in zebrafish. To test this hypothesis, we generated *edar*-knockout zebrafish using CRISPR/Cas9.

Compared to the wild-type (WT) fish with 11–14 gill rakers, *edar*$^{+/-}$ fish had fewer gill rakers with a forking morphology, and *edar*$^{-/-}$ fish lacked gill rakers (Fig. 3c). We measured the food intake of adult WT and *edar*$^{-/-}$ zebrafish (Fig. 3d, e). Compared with WT zebrafish (Fig. 3d), *edar*$^{-/-}$ fish had decreased food intake when fed with brine shrimp and increased food intake when fed with dead prey fish (Fig. 3e). The dead prey fish that we used were 12-days post-hatching (dph) zebrafish larvae with a length of 0.85–1.00 cm, exceeding the optimum length of one-third of the predator length.

We treated zebrafish larvae from 1–8 dph with EDAR activator (1 ng/ml activin A) or inhibitor (50 ng/ml BMP4). In control and inhibitor-treated fish, 5, 5, and 6 gill rakers were observed at 2, 3, and 8 dph, respectively. In fish treated with activin A, 5, 6, and 8 gill rakers were observed at 2, 3, and 8 dph, respectively (Fig. 4a). At 8 dph, the spacings of the gill rakers were 25.6 ± 1.6, 22.44 ± 0.66, and 30.7 ± 0.66 μm in control, activin A-treated and BMP4-treated zebrafish, respectively (Fig. 4d). Whole-mount in situ hybridization data showed that activin A treatment increased and BMP4 treatment decreased *edar* expression at 2, 3, and 8 dph (Fig. 4b; Supplementary Fig. 9).

Zebrafish larvae treated with activin A were able to feed on 1-day-old brine shrimp at 2 dph, whereas control fish and larvae treated with BMP4 were not (Fig. 4c). At 3 dph, larvae treated with activin A could feed on 2-day-old brine shrimp, but control and BMP4-treated larvae could feed on only 1-day-old brine shrimp (Fig. 4c). At 8 dph, more larvae in the activin A group were able to feed on 5-day-old brine shrimp than in the BMP4 and control groups (Fig. 4e).

**DHS mapping, reporter constructs, and luciferase assays.** Locus control regions (LCRs) are usually localized at DNase I hypersensitivity sites (DHS). Southern blot analysis revealed a DHS at the 5′ flanking region of *edar*, 13.7 kb upstream of the ATG translation initiation site (Fig. 5a, b). An Xvent-1 site was found in the DHS sequence. To investigate the DHS function, a luciferase reporter containing the DHS region, pGL6-p(−14,289/−13,455) (pGL6−1), and a shorter construct, pGL6-p(−13722/−13,455) (pGL6−2), which still contained the Xvent-1 site, were constructed (Fig. 5c; Supplementary Table 16). The transcriptional activities of pGL6−1 and pGL6−2 were both decreased compared with the activity of the pGL6 empty vector, which might be related to regulation by the Xvent-1 site ($P < 0.05$) (Fig. 5e, f). Sequence comparisons revealed that the four species of Sinipercidae had Xvent-1 sites at the 5′ flanking region of *edar* gene (Fig. 5d). We also quantified the *bmp4* copies in the adult *S. chuatsi* and zebrafish using RT-qPCR. Compared with zebrafish, *S. chuatsi* had significantly more copies of *bmp4* in both the gills and liver (Fig. 5g).

**Discussion**

SMRT sequencing enabled us to generate a high-quality, nearly complete genome assembly for *S. chuatsi*. Subsequently, the *S. chuatsi* genome assembly assisted in cost-efficient short-read assemblies for two closely related species, *S. kneri* and *S. scherzeri*, as well as the more distantly related species *C. whiteheadi*. Our approach of creating genome assemblies for whole families of organisms might serve as a roadmap for future and ongoing large-scale sequencing projects[23]. Assuming a continuous increase in the throughput and decrease in the cost of short- and long-read sequencing, a $1000 de novo-assembled teleost genome might be within reach.

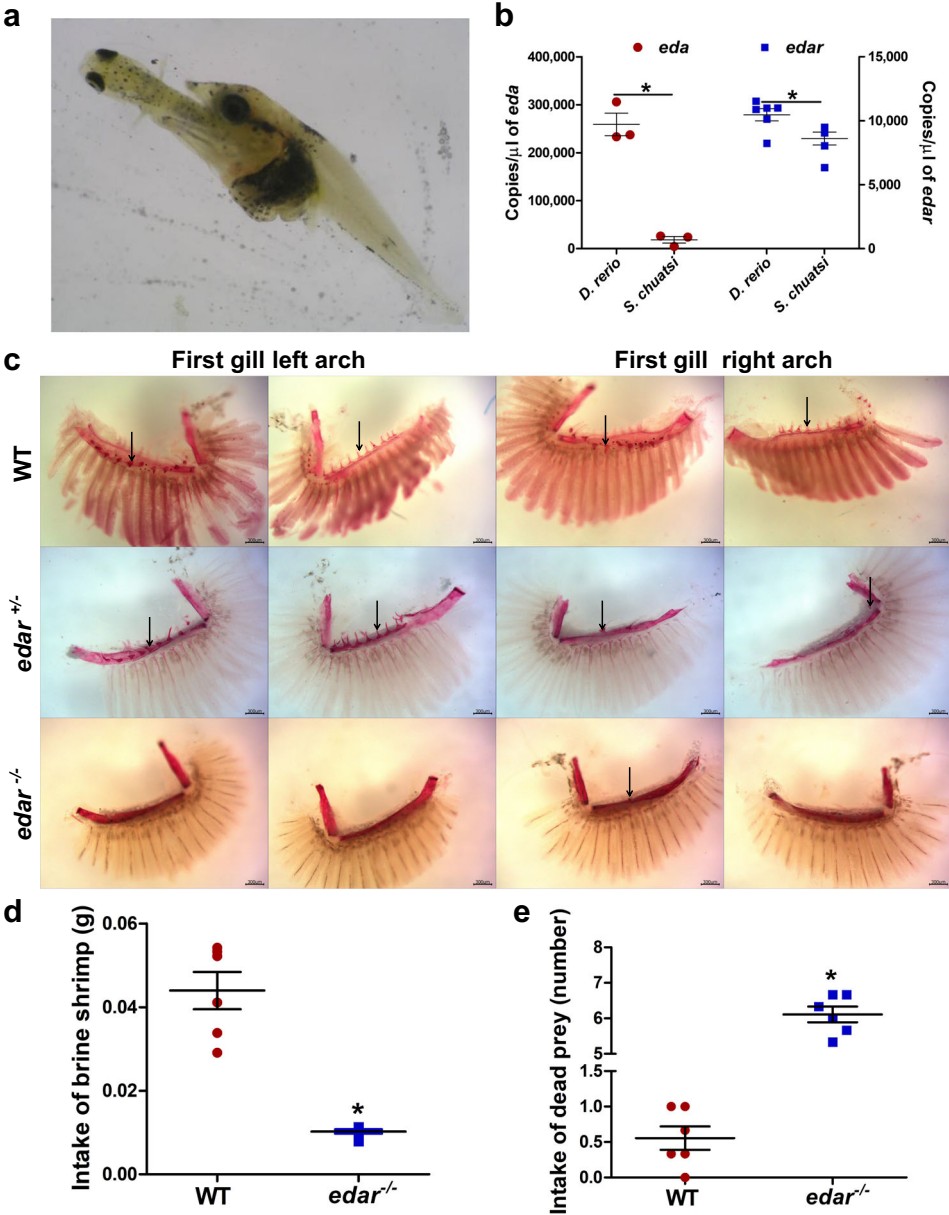

**Fig. 3 Gill arches, food intake, and absolute mRNA expression of *eda* and *edar*. a** *S. chuatsi* preying on a live fish. **b** Absolute copies of *eda* and *edar* in adult *S. chuatsi* and *D. rerio* (*n* = 3 for *eda*, *n* = 6 for *edar*). **c** The gill arches in adult zebrafish. WT adult zebrafish have 11–14 gill rakers; fewer gill rakers are observed in *edar*⁺/⁻ fish. In *edar*⁻/⁻ fish, 1–3 gill rakers remain on the outside of the first gill arch, whereas complete loss of gill rakers is observed on the inside of the first gill arch. **d**, **e** Intake of brine shrimp (**d**) and dead prey (**e**) by adult zebrafish (*n* = 6). The boxplot elements are defined as: center line, median; box limits, the third and first quartiles; whiskers, 1.5× interquartile range. * indicates a significant difference (*P* < 0.05).

We constructed a phylogenomic tree based on the concatenated noncoding fraction of the alignment. The highly supported resulting tree grouped the Sinipercidae into an independent monophylum. This special evolutionary position of the Sinipercidae is of great significance for studying the phylogenetic and ecological evolution of perch-like fish. The sinipercids are freshwater fishes endemic to East Asia, mainly distributed in Chinese river drainages, with a few species also found in Korea, Japan, and northern Vietnam[24]. They could hence represent a specialized group for biogeographic studies of the freshwater fish fauna in China. The common ancestor of all sinipercid fishes was estimated to have occurred 43.43 Mya, during the early Eocene. Our results are consistent with the time tree reported by Song et al.[25], which was based on protein-coding sequences of the Sinipercidae and calibrated by sinipercid fossil evidence. Here, the inferred time of the common ancestor of sinipercids was approximately 53.1 Mya (CI: 85.8–30.4 Mya). Our divergence time analysis provided a robust time estimate with a narrower CI for the Sinipercidae of ~43.43 Mya. At the larger time scale of the Tertiary and during the Eocene (54–34 Mya), warm and wet climatic conditions prevailed at northern latitudes, even within the Arctic, until 15 Mya, when the climate cooled progressively[26]. The climatic conditions of these geological periods were very favorable to sinipercid fishes. The results confirmed that the divergence among the Sinipercidae was rapid, especially for *S. chuatsi* and *S. kneri*.

Compared with *S. chuatsi* and *S. kneri*, *S. scherzeri* can easily be weaned onto dead prey fish or artificial diets[11]. Asparagine synthetase (*Asns*) deficiency leads to brain structural abnormalities and memory deficits[27]. *Asns* was positively selected on the

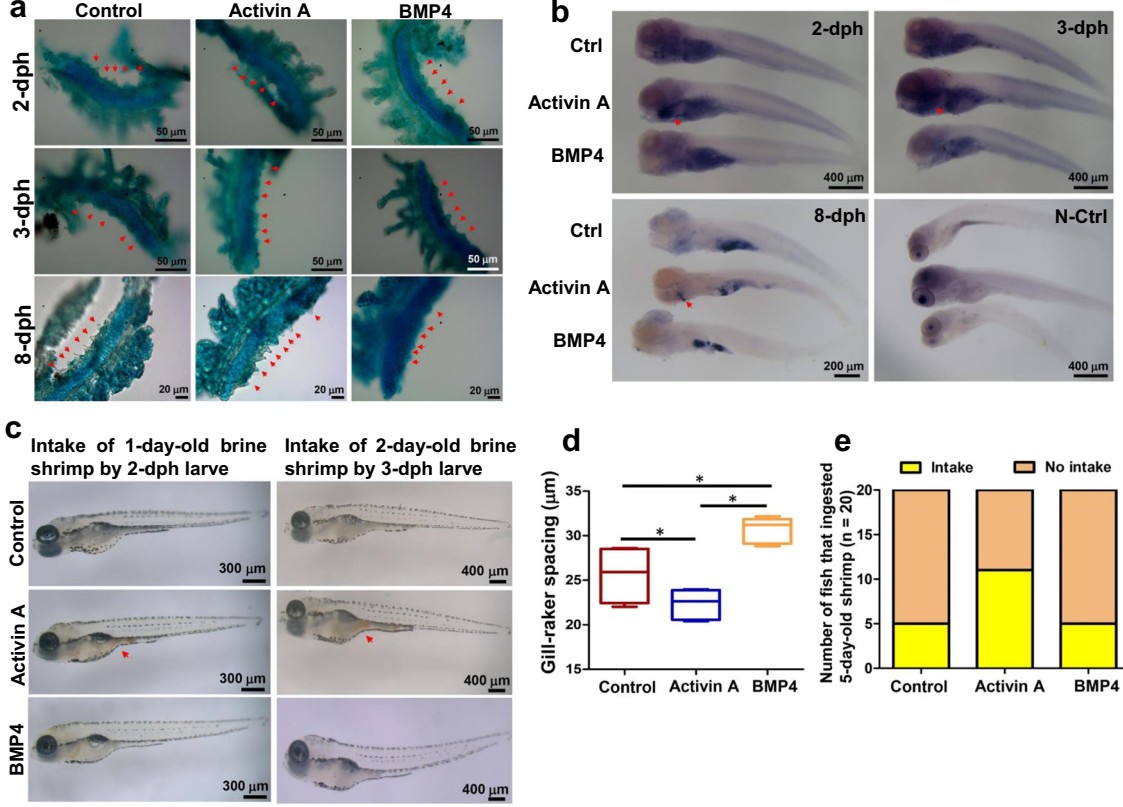

**Fig. 4 Gill rakers and food intake of zebrafish with activin A and BMP4 treatment. a** Number of zebrafish gill rakers (arrowheads, 0.1% alcian blue staining) at 2, 3 and 8 dph. **b** Zebrafish *edar* expression assessed by whole-mount in situ hybridization. Control fish or fish treated with 1 ng/ml activin A or 50 ng/ml BMP4 at 2, 3, 8 and dph and the negative control (N-Ctrl) are shown. **c** Zebrafish larvae intake of brine shrimp. **d** The distance between gill rakers of zebrafish at 8 dph ($n = 3$). **e** Zebrafish feeding at 8 dph. The numbers of fish that ingested 5-day-old shrimp were shown ($n = 20$). The boxplot elements are defined as: center line, median; box limits, the third and first quartiles; whiskers, 1.5× interquartile range. * indicates a significant difference ($P < 0.05$).

*S. scherzeri* branch, which was the easiest to wean onto artificial diets among the four species of mandarin fishes, and dead prey fish feeders had lower *asns* expression than nonfeeders, suggesting that the *Asns* gene might contribute to the specific feeding habit of mandarin fishes.

Although *S. kneri* shows the closest phylogenetic relationship with *S. chuatsi*, the growth rate of *S. chuatsi* is three to five times as high as that of *S. kneri*[12]. In the present study, we reported 411 PSGs with assigned gene symbols on the *S. kneri* or *S. chuatsi* branches. We searched the PSGs, which were differentially expressed in transcriptome data[22]. Altogether, 35 genes might be candidate genes for growth in mandarin fishes, including genes involved in immune regulation, skeletal muscle development, cell proliferation and differentiation, and reproduction and sexual dimorphism. For instance, the *galectin-3* gene in *S. chuatsi* was under positive selection and significantly upregulated in groups of fishes with higher growth performance, suggesting that protein glycosylation might contribute to the growth performance of *S. chuatsi*. Molecule interacting with CasL 2 (*Mical2*) is expressed in skeletal and cardiac muscles, and modulation of MICAL2 has an impact on skeletal muscle commitment[28]. *Mical2* was under positive selection in *S. kneri* and expressed at lower levels in slower-growing individuals, suggesting that MICAL2 might be involved in growth regulation. Previous studies reported that predator size and prey body mass had an important deterministic effects on the predator's feeding behavior, especially on the choice of prey[29]. The predatory feeding habit of *S. chuatsi* might have coevolved with its rapid growth.

Pyloric caeca secrete digestive enzymes and contain a large number of folds and microcilia, which can assist in absorption in fish. The numbers of pyloric caeca in *S. chuatsi, S. scherzeri*, and *S. kneri* are 117–323, 65–124, and 62–100, respectively. In contrast, *C. whiteheadi* has only 3 pyloric caeca[13]. In the present study, we identified 1687 PSGs in the *Coreoperca* branch and branch 5 leading to *Siniperca* sp. Among these PSGs, we found one gene cluster consisting of *mcm5*, *gcat*, *mchr1*, and *bptf* in *C. whiteheadi*. These genes have important roles in the origin of the germinal layer during embryo development, which might be related to the development of the pyloric caeca. *Bptf* deficiency decreases bone morphogenetic protein-4 (*Bmp4*) expression in the extraembryonic ectoderm in mice[30], BMP4 directly and indirectly acts on the epiblast to promote mesoderm and posterior identity[31]. Furthermore, we identified 8 genes involved in the development of the endoderm and the digestive tract that were under positive selection on branch 5 or the *Coreoperca* branch. Digestive enzyme activities are positively correlated with pyloric cecum number[32]. According to the different number of *pepsin A* genes between *Siniperca* sp. and *C. whiteheadi*, we inferred that the pseudogenization of one copy of the *pepsin A* gene might affect the function of pepsin, leading to the decrease in pyloric cecum number in *C. whiteheadi*.

The ancestors of Sinipercidae are marine fishes, which evolved in freshwater after repeated marine invasions and retreats in the early tertiary period in East Asia[24]. Before or during the Miocene, the last common ancestor of mandarin fishes occurred in the East China Sea, North Korea and Japan. However, the genetic basis of

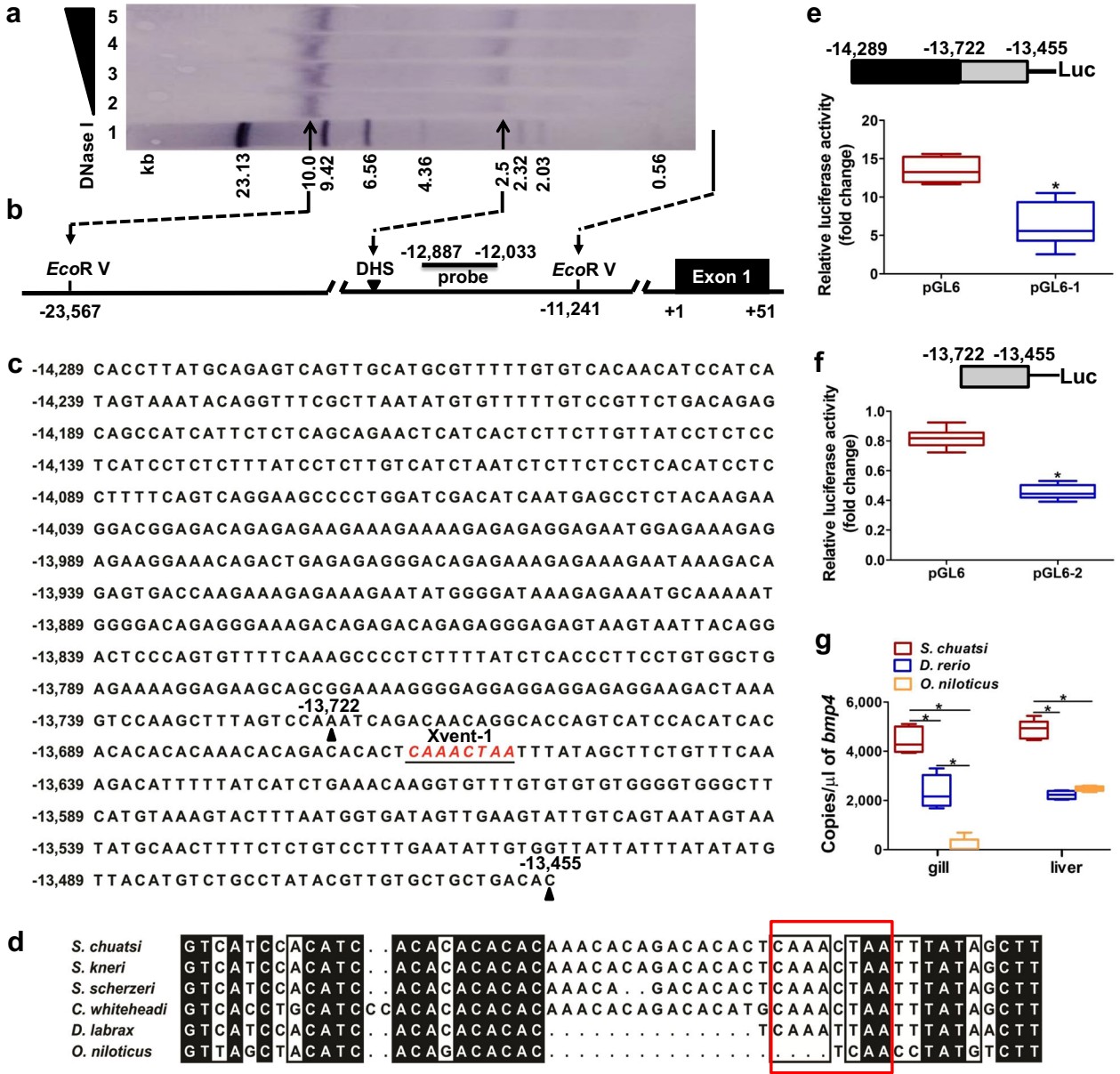

**Fig. 5 Identification and characterization of the DHS in the 5′ flanking region of *edar* in *S. chuatsi*. a** Identification of the DHS in the 5′ flanking region of *edar* in *S. chuatsi* by Southern blotting. **b** Schematic illustration of the DHS and *Eco*R V sites. +1 is the initiation codon ATG. **c** 5′ flanking region of *edar* in *S. chuatsi*. Nucleotides are numbered relative to the translation initiation site (ATG). **d** Alignment of the *edar* 5′ flanking region (partial sequence). The Xvent-1 site is highlighted by a red rectangle. **e** Luciferase activity of pGL6–1 ($n = 8$). **f** Luciferase assay for pGL6–2 ($n = 12$). **g** Absolute copies of *bmp4* in adult *S. chuatsi* ($n = 5$), *D. rerio* ($n = 4$), and *O. niloticus* ($n = 5$). The boxplot elements are defined as: center line, median; box limits, the third and first quartiles; whiskers, 1.5× interquartile range. * indicates a significant difference ($P < 0.05$).

salinity adaptation in mandarin fishes upon the transition from seawater to freshwater remains unclear. In the present study, we reported 127 PSGs involved in "ion transport" and 4 PSGs involved in "cellular hypotonic response". These results reflect that one of the key events in osmoregulation involves fine tuning of the ion-transporting apparatus of gill epithelia, which requires rearrangement of gill chloride cells and changes in Na$^+$/K$^+$-ATPase; Na$^+$, K$^+$, 2Cl$^-$ cotransporters of gills; and other critical ion transport proteins[33].

The Na$^+$/K$^+$-ATPase $\alpha-$1a isoform is a freshwater isoform involved in ion uptake, whereas the $\alpha-$1b isoform is a marine isoform associated with ion excretion[34]. The total number of copies of the Na$^+$/K$^+$-ATPase gene in mandarin fishes was more similar to that of marine and euryhaline fishes than to that of

freshwater fishes. The Na$^+$/K$^+$-ATPase $\alpha-$1a isoform of mandarin fishes was not duplicated as in freshwater fishes, which is consistent with its origin in marine environments. Hyperosmotic saline environments weaken chemical perception and processing, which are closely related to feeding behavior and detecting food[35]. The unique feeding behavior of mandarin fishes, which underwent an evolutionary transition from seawater to freshwater, is attributed to reductions in chemical sensing and enhancement of visual and lateral line sensing[8].

Ion regulation is tightly coupled to water flow across membranes, where aquaporins (AQPs) selectively facilitate the transport and exchange of water molecules. The expansion of *aqp* in European sea bass is associated with ion and water transport[36]. *Aqp1a, aqp3a, aqp3b,* and *aqp8b* were positively selected in

*L. calcarifer*, in *D. labrax* and on branch 7. Teleosts in seawater and brackish water, such as Asian sea bass, European sea bass and large yellow croaker, had three copies of *aqp8*, of which *aqp8aa* and *aqp8ab* were tandem duplicates and *aqp8b* originated from genome duplication. However, mandarin and cichlid fishes (*Maylandia zebra*, *Neolamprologus brichardi*, *Pundamilia nyererei*, *Haplochromis burtoni*, and *O. niloticus*), which thrive in freshwater and brackish water, had only two copies of *aqp8*. Therefore, marine sinipercids might have acclimated to freshwater by reducing water influx due to the absence of *aqp8aa*.

Predatory feeding behavior is a near-universal survival trait that is important for the acquisition of food[37]. Mandarin fishes show high predatory aggression in the wild[7]. In the present study, we found 5 PSGs on the *S. chuatsi* branch related to the MP term "abnormal aggression-related behavior", 87 PSGs in the *Siniperca* clade related to the MP term "abnormal emotion/affect behavior", and 4 genes (*pnmt*, *kirrel3*, *grn*, and *npas4*) with copy number variation between sinipercids and nonpredatory fish species. These genes might contribute to the predatory feeding behavior and predatory aggression of mandarin fishes.

Mice overexpressing *Pnmt*, the enzyme responsible for the conversion of noradrenaline to adrenaline, display increased home cage aggression[38]. Male mice lacking *Kirrel3*, a gene required for proper coalescence of vomeronasal organ axons into glomeruli in the accessory olfactory bulb, showed strongly reduced aggressive behavior in the resident-intruder paradigm[39]. Aggression was increased in $Grn^{-/-}$ and $Grn^{+/-}$ male mice[40] and in male mice lacking *Npas4*[41]. Our results indicated that the differential expression of these genes between feeders and nonfeeders, the expansion of *pnmt* and *kirrel3*, and the loss of *grn* and *npas4* might explain the predatory feeding behavior of mandarin fishes.

Gill rakers are bony or cartilaginous finger-like projections of gill arch, and have an important role in determining diet composition and prey size. Filter-feeding fishes have numerous elongated gill rakers[42], whereas omnivorous and carnivorous fishes have few short gill rakers[43]. Ectodysplasin-A (*eda*) and ectodysplasin-A receptor (*edar*) have been reported to control the development of gill rakers[44]. Using the gene expression of *edar* as a marker of raker primordia, the previous study identified quantitative trait loci on chromosomes 4 and 20 with large effects on gill raker reduction in stickleback fish, and indicated that the parallel developmental genetic features underlie the convergent evolution of gill raker reduction[45]. In this study, $edar^{-/-}$ zebrafish lacked gill rakers and had decreased food intake when fed with brine shrimp and increased food intake when fed with dead prey fish. At the same time, after the treatment of zebrafish larvae with EDAR activator, zebrafish larvae have more and denser gill rakers, and this treatment also has a significant effect on feeding of zebrafish larvae with brine shrimp. These results indicated that EDAR activation could accelerate gill raker development, and denser gill rakers might provide sufficient filtration of zooplankton. Therefore, the *edar* gene could be a key regulator of food selection and feeding habit in fish, and the lower expression of the *eda* and *edar* genes in *S. chuatsi* could contribute to its fewer gill rakers, resulting in the inability to feed on zooplankton and thus promoting the evolution of predatory feeding behavior.

LCRs are long-range *cis*-regulatory elements that regulate the expression of genes linked to physiological levels in a tissue-specific and copy number-dependent manner at ectopic chromatin sites[46]. An Xvent-1 site was identified in the DHS sequence in the 5′ flanking region of *edar*. BMP4 inhibits Xvent-1-mediated forkhead box A4 S homolog (FOXA4.S) transcriptional activation[47]. The four species of Sinipercidae, containing Xvent-1 sites in the *edar* gene, had 4–9 gill rakers. However, European sea bass and Nile tilapia did not contain the Xvent-1 site upstream of the *edar* gene and had >20 and >30 gill rakers, respectively[48,49], suggesting the number of gill rakers is related to the presence of the Xvent-1 site in the *edar* gene. The result of significantly more copies of *bmp4* in *S. chuatsi* indicated that BMP4-initiated signaling molecules, by binding to the Xvent-1 site in the *edar* gene, likely inhibit *edar* expression and gill raker development in *S. chuatsi*.

Furthermore, the genes upstream of several PSGs contained Xvent-1 sites, suggesting that these genes might be regulated by BMP4. For instance, alkylglycerol monooxygenase (*agmo*), *asns* and kinesin-like protein KIF20A isoform X3 (*kif20a*), which have been linked to feeding habit; galectin-3 and *mical2*, which are involved in growth regulation; *gcat* and *mchr1*, which are involved in the development of pyloric caeca; and aggression-related genes, such as *th*, *oxt*, *hnmt*, *kirrel3a*, and *pnmt*, also contained Xvent-1 sites. Teleosts, as the largest group of vertebrates, exhibit marked diversity in morphology, behavior and adaptation to various environments, especially through adaptive radiation, the process of rapid evolution from a common ancestor into an array of species[50]. Mandarin fishes have rapidly evolved, especially in terms of feeding and growth traits, which are closely related to the functional genes surrounding *eda* and *edar*. The identified LCR in the *edar* 5′ flanking region might affect the coordinated regulation of associated functional gene clusters through BMP4. The coevolution of these functional gene clusters might have an important role in the rapid adaptation of mandarin fishes to the environment.

In conclusion, we reported genome sequences of four members of the Sinipercidae. The results of positive selection analysis and genome comparisons revealed the adaptive evolution of genes related to predatory feeding, growth performance, pyloric cecum development, and euryhalinity. Mandarin fishes exclusively rely on live prey fish, possibly due to their low *edar* expression and very few gill rakers, leading to a predatory habit. These results contribute to a better understanding of the gene determination underlying predatory feeding behavior in mammals and the coevolution of feeding, growth, digestion, and euryhalinity traits in teleosts.

## Methods

**Ethics statement.** The study was approved by the Institutional Animal Care and Use Ethics Committee of Huazhong Agricultural University (Wuhan, Hubei, China). The protocol was approved by the committee on the ethics of animal experiments of the university. All efforts were made to minimize suffering in animals.

**Fish materials and sequencing.** Inbred lines of the mandarin fish species *S. chuatsi*, *S. kneri*, *S. scherzeri* and *C. whiteheadi* were obtained from the Chinese Perch Research Centre of Huazhong Agricultural University. Genomic DNA was extracted from the muscle of *S. chuatsi* (female), *S. kneri* (male), *S. scherzeri* (male), and *C. whiteheadi* (male) with a standard phenol-chloroform method. The diploid adult female *S. chuatsi* genome was sequenced on a PacBio RS II platform using P6-C4 chemistry (Pacific Biosciences, Menlo Park, CA, USA; 50× raw-read coverage). The transcriptome libraries of *S. chuatsi* brain, muscle, liver and gut tissues were prepared and sequenced using the Illumina HiSeq 4000 system (HiSeq 4000 SBS Kit, paired-end). The diploid adult male *S. kneri*, *S. scherzeri* and *C. whiteheadi* libraries for genome resequencing were prepared with the standard procedures for 300-bp paired-end libraries, and additional libraries were constructed using a Nextera® Mate-Pair Library Preparation Kit (Illumina, San Diego, CA, USA). Libraries were sequenced on an Illumina HiSeq 4000 system. Library construction and sequencing were performed by the Beijing Genomics Institute (BGI, Wuhan, Hubei, China).

**Genome assembly of *S. chuatsi* from SMRT long-sequence reads.** We ran the Canu v1.0 de novo assembly pipeline[17] (including correction, trimming and assembly steps) using the SMRT long-read data of *S. chuatsi*. A few parameters were adapted to the server used to perform the calculations (Dell PowerEdge R910; 80 Threads E7–4870@2.40 GHz; RAM: 1031.8 GB):

maxMemory=500, maxThreads=80, genomeSize=800000000, and errorRate=0.06.

A second de novo assembly of the SMRT data was performed using the FALCON assembler[16]. We did not use FALCON read correction but directly applied FALCON to assemble the corrected reads from the Canu assembler, saving considerable computing time. The FALCON assembly was optimized for an N50 contig length by varying the "length_cutoff_pr" parameter.

The Canu assembly was the most contiguous assembly but also revealed more potential misassemblies, as depicted by interchromosomal fusions when aligning it to related species genome assemblies using the LAST aligner[51]. We found that it was possible to improve Canu assembly continuity by aligning the assembly to the FALCON assembly and inferring scaffolds by Ragout[52]. Subsequently, potential misassemblies in scaffolds were removed by splitting them at locations that could not be verified to be collinear with the alternative assembly (FALCON) or genome assemblies of related species (Asian sea bass[53], European sea bass[36], or three-spined stickleback[54]). The polished assembly of *S. chuatsi* was ordered into 24 chromosomal groups by alignment (LAST) with the Asian sea bass genome and resolving the breakpoint graph using Ragout. Finally, the consensus sequence was corrected using the Quiver algorithm[55], resulting in the first chromosomal-scale genome assembly, named "sinChu4".

**Further improvements to the genome assembly.** Comparison with a high-density genetic linkage map of *S. chuatsi* showed only a few discrepancies with the first chromosomal-scale genome assembly of *S. chuatsi* (sinChu4). A second round of genome improvement was carried out by mapping all SMRT raw reads back to the genome (using BWA-MEM[56], with SMRT read-specific parameters). Contigs/scaffolds were split at locations in the assembly that were not covered by at least one pair of SMRT reads with a min. overlap of 100 bp and a mapping score of 60.

Subsequently, using SMRT raw reads connecting contigs/scaffolds allowed us to markedly improve assembly contiguity. If chromosomal synteny was not violated, we allowed contig/scaffold links inferred by a minimum of one SMRT read. Long-read scaffolding was implemented by custom scripts involving LAST aligner for fast raw SMRT read mapping as well as SSPACE[57] for scaffolding.

The resulting large scaffolds were placed according to the first assembly of *S. chuatsi* and the genetic linkage map (unpublished). Some yet-unordered pieces of the assembly could be placed manually into the chromosomal assembly according to the genetic linkage map.

Finally, several gap closure methods were applied. First, we used PBJelly[58] to extend or close gaps by SMRT raw reads. Second, we joined many neighboring contigs in the scaffolds that exhibited large overlaps between contigN ends and contigN+1 starts. These overlaps were identified by BLASTn[59] and joined using custom scripts. The genome assembly consensus sequences were corrected again after these improvements by using the Quiver algorithm, with the resulting assembly called "sinChu7". The methods for short-read genome assembly of *S. kneri*, *S. scherzeri* and *C. whiteheadi* are described in the Supplementary Information.

**Phylogenomic tree construction.** RAxML version 8.2.4[19] was used to construct a phylogenetic species tree (fast topology search using the GTRCAT evolutionary model and parameter -m GTRCAT -f E, followed by nearest-neighbor-interchange (NNI) optimization under the GTRGAMMAI model and SH-Test: -m GTRGAMMAI -F J), based on the noncoding fraction of whole-genome sequence alignments. Tree calculations for Percomorphaceae subsets were performed by more exhaustive calculations using RAxML-NG[60] and IQTREE2[61], which both delivered congruent results. Furthermore, a chronogram of the fishes was calibrated using MCMCTree[62].

**Analysis of positive selection.** Coding sequences for each species were reconstructed from 1-to-1-ortholog whole-genome alignments using *S. chuatsi* coding sequence annotations as a reference. Coding sequences for each gene and all species were combined in a multiple FASTA file. For each gene, two multiple-codon alignments were produced using different alignment methods (MACSE[63] and TranslatorX[64]). Each codon alignment was tested twice (allowing gaps/not allowing gaps in the alignments) for positive selection on all branches of the phylogenetic tree using the aBSREL method in the HyPhy package[65]. Genes were considered positively selected if an uncorrected FDR $P < 0.05$ was detected in three of the four tests. To be conservative, the largest $P$-value of the three was assigned to the PSG. Enrichment tests were performed at www.mousemine.org. All annotated genes with assigned gene symbols in *S. chuatsi* were used as the background gene list for enrichment tests. Paralogs in fishes, which are mainly due to teleost-specific whole-genome duplication, needed to be treated as single-copy genes in this regard. For instance, *Cyp19a1a* and *Cyp19a1b* were both treated as *Cyp19a1*.

**Gill rakers and food intake of *edar*$^{-/-}$ zebrafish.** Zebrafish larvae (AB strains) were obtained from the China Zebrafish Resource Center (CZRC, Wuhan, China). CRISPR gRNAs against the zebrafish *edar* gene were designed with the online tool ZiFiT (http://zifit.partners.org/ZiFiT/CSquare9Nuclease.aspx) (Supplementary Table 17). gRNAs were generated with a pMD19-T vector template (TaKaRa, Dalian, China) and transcribed using a Transcript Aid T7 High-Yield Transcription Kit (Thermo Fisher Scientific, Waltham, MA, USA). One-cell-stage WT embryos were coinjected with 300 ng/μl Cas9 mRNA and 30 ng/μl gRNA. F1 heterozygotes were generated through outbreeding between an F0 male and a WT female, and *edar*$^{-/-}$ mutants (F2 homozygous) were obtained by inbreeding F1 heterozygote mutant strains (Supplementary Fig. 10). A nested PCR assay was conducted to identify CRISPR-induced mutations (Supplementary Table 17).

Gill rakers of adult WT and *edar*$^{-/-}$ zebrafish were stained with 0.008% alizarin red S (Sinopharm Chemical Reagent, Shanghai, China) as previously described and then photographed by a stereomicroscope (Olympus, Tokyo, Japan) equipped with a digital camera. Zebrafish were fed equal amounts of brine shrimp (*Artemia* nauplii) or dead zebrafish larvae (total length: 0.85–1.00 cm) and then weighed before and after feeding ($n = 6$); alternatively, the number of remaining dead prey fish was considered ($n = 6$).

**Absolute mRNA expression of *eda* and *edar*.** RNA was extracted from gill rakers of *S. chuatsi* and zebrafish and reverse-transcribed into cDNA. Absolute RT-qPCR was performed for each sample with three technical replicates using AceQ® qPCR SYBR® Green Master Mix (Vazyme, Nanjing, China) on a MyiQ™ 2 Two-Color Real-Time PCR Detection System (Bio-Rad, Hercules, CA, USA) (primers listed in Supplementary Table 18). The logarithms of the copy numbers of *eda* and *edar* were calculated from the standard curve.

**Treatment of activator and inhibitor of *edar* in zebrafish.** Zebrafish larvae treated with the *edar* activator activin A (R&D Systems, Minneapolis, MN, USA) at a final concentration of 1 ng/ml or the *edar* inhibitor BMP4 (R&D Systems) at 50 ng/ml at 8 dph were compared to fish without treatment as a control. For gill raker staining, the larvae were stained with 0.1% alcian blue (Sinopharm Chemical Reagent Co.) as previously described. Gill arches were photographed by a stereo-microscope, and the spacings of the gill rakers ($n = 3$) were calculated with CS Illustrator 6.0 (Adobe, San Jose, CA, USA). Whole-mount in situ RNA hybridization was performed as previously described[66]. Total RNA was extracted from zebrafish larvae and reverse-transcribed using M-MLV reverse transcriptase (Takara) according to the manufacturer's instructions. Partial cDNA of *edar* was amplified, then cloned into the pGEM-T Easy vector (Promega) (Addgene ID 149381, Supplementary Table 17). DIG-labeled RNA probes were synthesized by in vitro translation after plasmid linearization using the T7 or SP6 RNA Polymerase (Promega). Food intake of the control zebrafish and zebrafish treated with activin A and BMP4 was recorded at 9:00 and 18:00 every day. In each group, larvae were randomly divided into three tanks ($n = 20$) and fed brine shrimp of different sizes.

**DHS mapping.** Nucleus isolation was performed with blood samples of adult *S. chuatsi* according to a previously described procedure[67]. The nuclei were treated with 1.25, 2.5, 3.75, and 5 U/ml DNase I (TaKaRa). Then, genomic DNA (30 μg) was purified for southern blot analysis after *Eco*R V (TaKaRa, 30 U) digestion according to the supplier's instructions. Probes were prepared by PCR amplification using the following primers: Probe-F 5′-AGATGGCGGAGTTGAGTTG TAT-3′ and Probe-R 5′- TTGTCTTCCACCTTCCAACCTT-3′.

Xvent-1, a transcription factor binding site within the *edar* 5′ flanking region targeted by the DHS map (Fig. 5a), was identified using Web Promoter Scan Service (https://www-bimas.cit.nih.gov/molbio/proscan/). Two genomic fragments containing the Xvent-1 site (Supplementary Table 16) were synthesized by GenScript (Nanjing, Jiangsu, China) and cloned into the luciferase reporter vector pGL6 (Beyotime, Shanghai, China). HEK293T cells were plated at a density of $2.5 \times 10^5$ cells per well of a 24-well plate and transfected with combinations of plasmids pGL6–1 (Addgene ID 149382, 500 ng/well) and pGL6–2 (Addgene ID 149383, 500 ng/well), respectively, as previously described[68]. All experiments were carried out four times (independent transfections). In addition, the copy numbers of *bmp4* in the gills and livers of adult *S. chuatsi*, *D. rerio*, and *O. niloticus* were calculated as mentioned above.

**Statistics and reproducibility.** Statistical analysis was performed using GraphPad Prism 5 software. All data were obtained from at least three independent experiments. In the in vitro studies, unpaired *t*-test between two groups or one-way ANOVA among multiple groups were used to calculate *P*-values. The *P*-values for functional enrichment of PSGs were calculated using the Hypergeometric distribution at www.mousemine.org.

**Reporting summary.** Further information on research design is available in the Nature Research Reporting Summary linked to this article.

## Data availability

We have set up a genome browser (UCSC type) for the genome assemblies of *Siniperca chuatsi*, *Siniperca scherzeri*, *Coreoperca whiteheadi*, and *Siniperca kneri*, accessible at http://genomes.igb-berlin.de. The raw sequencing data and genome assemblies are publicly available at NCBI under accession number PRJNA513951. Source data underlying plots are in Source Data. All other data and/or materials are available upon reasonable request from X.-F.L. and/or H.K.

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

## Acknowledgements

This project was supported by the China Agriculture Research System (CARS-46 to X.-F.L.), the German Research Foundation (DFG KU3596/1-1 to H.K.), and the National Natural Science Foundation of China (31772822 and 31272641 to X.-F.L. and 31602131 to S.H.).

## Author contributions

X.-F.L., S.H., and H.K. conceived this genome project and coordinated research activities; X.-F.L. and S.H. performed genome sequencing; H.K. performed genome assembly and bioinformatics analyses; S.H., L.L., L.-Y.L., W.-J.C., Y.-Q.D., J.L., J.X., and Y.-P.Z. performed comparative genomic analyses; L.-Y.L., S.-L.T., X.C., Z.Z., Z.Y., and L.L. performed the analyses of gill rakers and the function and regulation of *edar*; S.H., H.K., and L.L. wrote the manuscript; and Y.-X.T., S.H., H.K., X.-F.L., and S.W. revised the manuscript.

## Competing interests

The authors declare no competing interests.
