## [Peer Review File · Communications Biology]

Reviewers' comments:

Reviewer #1 (Remarks to the Author):

The authors performed genome sequencing of several mandarin fishes (including *Siniperca chuatsi*, *S. kneri*, *S. scherzeri* and *Coreoperca whiteheadi*), and generated an *edar*^{-/-} zebrafish variant with a "gill raker-loss" phenotype to provide novel insights into the innate predatory feeding behavior. Generally speaking, the work produced good data and proposed an interesting hypothesis. However, the authors are recommended to make major revisions before acceptance for publication.

Major issues:

1. There are too many grammatical errors. The authors are therefore recommended to obtain editing service from a professional company or a native English speaker. For example, "was" should be "were" on line 169, "and" should be corrected as "that" on line 170; "generate" should be changed to "generated" on line 204; addition of "using" before "" fish without" on line 224; "arch" should be "arches" on line 226, since "were" was used; "was" should be revised as "were" on line 227, since the subject is "spacings"; "genome wide" should be "genome-wide" on line 364.
2. It is unreasonable to assemble polished genome of *Siniperca chuatsi* in a 24-chromosomal format using the published Asian sea bass genome as the reference (on lines 142-144), since these two fishes are far away phylogenetically from each other.
3. Please provide more discussions on lines 392-394, since the conclusion is not solid (lines 593-594 either).

Minor issues:

1. Many format issues should be resolved. For example, the cited reference style should be consistent as "xxx et al, year". Please revise it at least on lines 94-95, 235, 262, 272, 290, 313, 328, 441, 448-449, 469-470, 477-479, 521, 523, 532-535, 542, 576, 586, and 590.
2. Please provide the full name for any abbreviated term when it appeared at the first time in the text, such as FDR and PSG on lines 189-190, *edar* and WT under the section 2.6, BMP4 in the subtitle of section 2.7., and Mya on line 324.
3. References for applied software or previously reported data should be provided, such as "the Hyphy package" on lines 188-189 and the established linkage map on lines 276-277.
4. Please rewrite the sentence on lines 299-301.
5. Figure 1 should be simplified, since most topologies of the phylogenetic tree were reported before.
6. Figure 3b should be enlarged for a better vision.

Reviewer #2 (Remarks to the Author):

Given the economical value and special first feeding behaviour of mandarin fish, it is necessary to identify the genetic architecture underlying the formation of feeding behavior through the genomic sequencing. The authors reported a well chromosomal level genome of *S. chuatsi* and their three relative's genomes. They found many genes related to predatory feeding, growth performance etc using positive selection analysis. Interestingly, they also found that the Xvent-1 binding site upstream of *edar* gene which is related to the phenotype of gill raker-loss. Anyway, the interpretation of the results is sound for the most part, and gives enough proof to verify their results. Nevertheless, some conclusions would be strengthened with some reorganization and more thorough editing by a native English reader with some expertise in the science presented.

I just have few concerns as followed:

1. In line 278-281, the authors said more than 99.82% genetic markers were used for the construction of pseudochromosome. I think here the important thing is the relationship between the order of genetic markers and scaffolds. Also how many scaffolds can be orientated in the pseudochromosome based on the genetic map?
2. The authors performed the gene annotation by RNA-seq and homology predication. How about the denovo prediction?
3. In line 343-345, the authors should provide the number of chromosomal rearrangements which is consistent with the evolutionary distance.
4. The main story of this manuscript is the genetic of feeding behavior. I don't know how much the

results of growth analysis and salinity adaptation have to do with feeding behavior. It should be point out the relationship clearly.

5. Almost all genomic analyses start with positive selected genes. But for the part of function of edar, the most important part in this manuscript, it looks independent of the overall genomic analysis and thus it leads to poorer integrity of the genome article. Are they also positive selected genes?

Reviewers' comments:

Reviewer #1 (Remarks to the Author):

The authors performed genome sequencing of several mandarin fishes (including *Siniperca chuatsi*, *S. kneri*, *S. scherzeri* and *Coreoperca whiteheadi*), and generated an *edar*^{-/-} zebrafish variant with a “gill raker-loss” phenotype to provide novel insights into the innate predatory feeding behavior. Generally speaking, the work produced good data and proposed an interesting hypothesis. However, the authors are recommended to make major revisions before acceptance for publication.

Major issues:

1. There are too many grammatical errors. The authors are therefore recommended to obtain editing service from a professional company or a native English speaker. For example, “was” should be “were” on line 169, “and” should be corrected as “that” on line 170; “generate” should be changed to “generated” on line 204; addition of “using” before “” fish without” on line 224; “arch” should be “arches” on line 226, since “were” was used; “was should be revised as “were” on line 227, since the subject is “spacings”; “genome wide” should be “genome-wide” on line 364.

Reply:

We revised the grammatical errors as suggested (“was” was revised as “were” on **Line 512**; “and” was revised as “with the resulting assembly called “sinChu7” on **Line 513**; “generate” was revised as “generated” on **Line 546**; “fish without treatment” was revised as “compared to fish without treatment as a control” on **Line 566**; “arch” was revised as “arches” on **Line 568**; “was” was revised as “were” on **Line 569**; “genome wide” was revised as “genome-wide” on **Line 197**).

We also used the editing service provided by the professional language editing company (Nature Research Editing Service: <https://authorservices.springernature.com/language-editing/>) suggested in the submission guidelines of Communications Biology.

This document certifies that the manuscript
Mandarin fish (Sinipercidae) genomes provide insights into innate predatory feeding

prepared by the authors

Shan He^{1,5}, Ling Li^{1,5}, Li-Yuan Lv^{1,5}, Wen-Jing Cai¹, Ya-Qi Dou¹, Jiao Li¹, Shu-Lin Tang¹, Xu Chen¹, Zhen Zhang¹, Jing Xu¹, Yan-Peng Zhang¹,...

was edited for proper English language, grammar, punctuation, spelling, and overall style by one or more of the highly qualified native English speaking editors at SNAS.

This certificate was issued on **March 4, 2020** and may be verified on the SNAS website using the verification code **21AB-A36A-CF32-E88F-168P**.

Neither the research content nor the authors' intentions were altered in any way during the editing process. Documents receiving this certification should be English-ready for publication; however, the author has the ability to accept or reject our suggestions and changes. To verify the final SNAS edited version, please visit our verification page at secure.authorservices.springernature.com/certificate/verify.

If you have any questions or concerns about this edited document, please contact SNAS at support@as.springernature.com.

SNAS provides a range of editing, translation, and manuscript services for researchers and publishers around the world. For more information about our company, services, and partner discounts, please visit authorservices.springernature.com.

2. It is unreasonable to assemble polished genome of *Siniperca chuatsi* in a 24-chromosomal format using the published Asian sea bass genome as the reference (on lines 142-144), since these two fishes are far away phylogenetically from each other.

Reply:

As of the beginning of 2020, several new genome assemblies are available that would be more suitable than the Asian sea bass assembly for reference-assisted scaffolding. However, these assemblies were not available to us in 2017. At that time, the Asian sea bass genome was the highest-quality genome (actually, the first long-read fish genome assembly) with a suitable divergence time from *S. chuatsi*. As we state in the manuscript, we used Asian sea bass sequences only to infer contig order, not to build the contigs. Contig assembly was performed using *de novo* approaches (Canu/Falcon). Contig order was also not inferred purely by synteny with the Asian sea bass, as this information was used only if at least one *S. chuatsi* long read supported the contig order and orientation. In this way, we were able to make better use of the information in the long-

read data. Without this approach, the chromosomal-level assembly of *S. chuatsi* would have been much less complete.

In case any doubts remain, we recently generated Hi-C data for *S. chuatsi* and reassembled the genome. The assembly from 2017 showed only a few small differences in dot plots and thus is in fact state-of-the-art. Our brand-new Hi-C-based chromosomal assembly represents data not used in the current manuscript; thus, we attach a dot plot comparison for the reviewers (see below).

Inspired by the finding that synteny with diverged genomes can greatly facilitate chromosome-scale assembly, we have recently designed a new bioinformatics pipeline, and the corresponding publication is currently under review. This software pipeline is already freely available at <https://github.com/HMPNK/CSA2.6>.

3. Please provide more discussions on lines 392-394, since the conclusion is not solid (lines 593-594 either).

Reply:

Considering the contextual relationships, we rewrote the results and discussion related to the Xvent-1 site: "...Furthermore, the genes upstream of several PSGs contained Xvent-1 sites, suggesting that these genes might be regulated by BMP4. For instance, alkylglycerol monooxygenase (*agmo*), *asns* and kinesin-like protein KIF20A isoform X3 (*kif20a*), which have been linked to feeding habit; *galectin-3* and *mical2*, which are involved in growth regulation; *gcat* and *mchr1*, which are involved in the development of pyloric caeca; and aggression-related genes, such as *th*, *oxt*, *hnmt*, *kirrel3a* and *pnmt*, also contained Xvent-1 sites. ...” (Lines 427-432)

Minor issues:

1. Many format issues should be resolved. For example, the cited reference style should be consistent as “xxx et al, year”. Please revise it at least on lines 94-95, 235, 262, 272, 290, 313, 328, 441, 448-449, 469-470, 477-479, 521, 523, 532-535, 542, 576, 586, and 590.

Reply:

We revised the cited reference style as suggested in the submission guidelines of Communications Biology.

2. Please provide the full name for any abbreviated term when it appeared at the first time in the text, such as FDR and PSG on lines 189-190, *edar* and WT under the section 2.6, BMP4 in the subtitle of section 2.7., and *Mya* on line 324.

Reply:

We added the full name for abbreviations as suggested (Lines 158, 190, 193, 206, 216, 246, 261, 266-269, 271-272, 276-277, 336-338, 343-348, 368-369, 382, 386-387, 428-429).

3. References for applied software or previously reported data should be provided, such as “the HyPhy package” on lines 188-189 and the established linkage map on lines 276-277.

Reply:

We added a reference for the HyPhy package (Reference 92, Lines 531, 806-807):

Pond, S. L., Frost, S. D. & Muse, S. V. HyPhy: hypothesis testing using phylogenies. *Bioinformatics* 21, 676–679 (2005).

The linkage map is unpublished (Line 506), may be provided upon request by Shan He, HZAU, Wuhan China.

4. Please rewrite the sentence on lines 299-301.

Reply:

We rewrote the sentence as follows:

“... As undefined insert sizes may result in less efficient scaffolding during genome assembly, we mapped these reads to the high-quality *S. chuatsi* reference genome and sorted mapped mate pairs of a particular insert size range into distinct in silico size-selected files. ...” (Lines 133-135).

5. Figure 1 should be simplified, since most topologies of the phylogenetic tree were reported before.

Reply:

Fig. 1 has been replaced by a new tree with fewer species (Line 844) and the old one was placed into Supporting Information (Fig. S3).

6. Figure 3b should be enlarged for a better vision.

Reply:

We merged the 2 bar charts in Figure 3b and enlarged it as suggested.

Reviewer #2 (Remarks to the Author):

Given the economical value and special first feeding behaviour of mandarin fish, it is necessary to identify the genetic architecture underlying the formation of feeding behavior through the genomic sequencing. The authors reported a well chromosomal level genome of *S. chuatsi* and their three relative's genomes. They found many genes related to predatory feeding, growth performance etc using positive selection analysis. Interestingly, they also found that the Xvent-1 binding site upstream of *edar* gene which is related to the phenotype of gill raker-loss. Anyway, the interpretation of the results is sound for the most part, and gives enough proof to verify their results. Nevertheless, some conclusions would be strengthened with some reorganization and more thorough editing by a native English reader with some expertise in the science presented.

I just have few concerns as followed:

1. In line 278-281, the authors said more than 99.82% genetic markers were used for the construction of pseudochromosome. I think here the important thing is the relationship between the order of genetic markers and scaffolds. Also how many scaffolds can be orientated in the pseudochromosome based on the genetic map?

Reply:

We did not state that we used all markers; we simply say that most markers of a single linkage group also mapped to a distinct chromosomal sequence in our final assembly.

Indeed, only a fraction of the markers were useful for anchoring contigs/scaffolds. As already stated in the manuscript, the successful chromosomal assembly heavily relied on having large scaffolds before ordering them by the linkage map. Ordering hundreds or thousands of smaller scaffolds by genetic linkage maps would be error prone.

We obtained these large scaffolds using our combined synteny/long-read strategy, as described above in response to reviewer #1. We ordered 76 scaffolds using the linkage map. Because there were only a few scaffolds per chromosome (1 - 8 scf/chr), ordering them by the linkage map was straightforward and could be done manually without sophisticated software. These numbers are now presented in the manuscript in **Lines 114-115**.

The chromosomal assembly results are highly consistent with a Hi-C assembly that we performed recently (see the dot plot in response to reviewer #1) and underline the validity of our approach.

2. The authors performed the gene annotation by RNA-seq and homology predication. How about the de novo prediction?

Reply:

We did not rely on *de novo* or *ab initio* prediction for several reasons.

Compared to gene models inferred from protein homology, *ab initio* models are highly error prone. The probability of predicting a correct exon is only approximately 90%, even if its posterior probability is close to 100% (see, for example, Table 1 at <https://www.ncbi.nlm.nih.gov/pmc/articles/PMC1538822/>). The average teleost gene has 9-10 exons; thus, the probability of having suboptimal *ab initio* gene models is somewhat high. Such

errors could result in strong biases in downstream analyses, such as positive selection analysis, which we performed in this manuscript.

In our opinion, *ab initio* gene prediction is warranted only if little or no homologous protein or RNAseq data are available for a new organism. In fishes, this is not the case. For example, we used approximately 1,700,000 teleost protein sequences for the annotation of *S. chuatsi*, and even more importantly, we used several RNAseq datasets of different tissues for annotation. Annotation pipelines frequently miss some genes, but as already stated in the manuscript, the BUSCO scoring of our genome assembly and our genome annotation shows that the annotation is not missing too many genes and is highly complete (C: well above 94%, M: below 2%):

sinChu7_genome: C: 97.1% [S: 94.7%, D: 2.4%], F: 1.4%, M: 1.5%;

SC7_annotation: C: 94.5% [S: 91.0%, D: 3.5%], F: 3.7%, M: 1.8%;

C=complete, S=single, D=duplicated, F=partial, M=missing

3. In line 343-345, the authors should provide the number of chromosomal rearrangements which is consistent with the evolutionary distance.

Reply:

We added the requested information, "... (*S. kneri* = 79; *S. scherzeri* = 76; *C. whiteheadi* = 162; *D. labrax* = 251; *L. calcarifer* = 345)", to the end of the sentence (Lines 178-179).

4. The main story of this manuscript is the genetic of feeding behavior. I don't know how much the results of growth analysis and salinity adaptation have to do with feeding behavior. It should be point out the relationship clearly.

Reply:

We added some discussion of the relationship between growth and feeding behavior as follows:

"... Previous studies reported that predator size and prey body mass had an important deterministic effects on the predator's feeding behavior, especially on the choice of prey (Meers, 2002; Scharf et al., 2000). The predatory feeding habit of *S. chuatsi* might have coevolved with its rapid growth. ..." (Lines 250-252)

We also added some discussion of the relationship between salinity adaptation and feeding behavior as follows:

“... Hyperosmotic saline environments weaken chemical perception and processing, which are closely related to feeding behavior and detecting food (Dole et al., 1985). The unique feeding behavior of mandarin fishes, which underwent an evolutionary transition from seawater to freshwater, is attributed to reductions in chemical sensing and enhancement of visual and lateral line sensing (Liang et al., 1998). ...” (Lines 308-312)

References:

- (1) Meers, M. B. Maximum bite force and prey size of *Tyrannosaurus rex* and their relationships to the inference of feeding behavior. *Hist. Biol.* **16**, 1-12 (2002).
- (2) Scharf, F. S., Juanes, F. & Rountree, R. A. Predator size-prey size relationships of marine fish predators: interspecific variation and effects of ontogeny and body size on trophic-niche breadth. *Mar. Ecol. Prog. Ser.* **208**, 229-248 (2000).
- (3) Dole, J. W., Rose, B. B. & Baxter, C. F. Hyperosmotic saline environment alters feeding behavior in the western toad, *Bufo boreas*. *Copeia* **1985**, 645-648 (1985).
- (4) Liang, X. F., Liu, J. K. & Huang, B. Y. The role of sense organs in the feeding behaviour of Chinese perch. *J. Fish Biol.* **52**, 1058-1067 (1998).

5. Almost all genomic analyses start with positive selected genes. But for the part of function of *edar*, the most important part in this manuscript, it looks independent of the overall genomic analysis and thus it leads to poorer integrity of the genome article. Are they also positive selected genes?

Reply:

We rewrote this part (Lines 363-366, 368-373). Gill rakers are bony or cartilaginous finger-like projections of gill arch (Sanderson et al., 2001), and play an important role in determining diet composition and prey size (Magnuson and Heitz. 1971). Filter-feeding fishes have numerous elongated gill rakers (Roesch et al., 2013), while omnivorous and carnivorous fishes have few short gill rakers (Abumandour and El-Bakary, 2017). Ectodysplasin-A (*eda*) and ectodysplasin-A receptor (*edar*) have been reported to control the development of gill rakers (Harris et al., 2008). Using the gene expression of *edar* as a marker of raker primordia, the previous study identified quantitative trait loci on chromosomes 4 and 20 with large effects on gill raker reduction in

stickleback fish, and indicated that the parallel developmental genetic features underlie the convergent evolution of gill raker reduction (Glazer et al., 2014). The mandarin fish genomes enabled us to explain the reductions in gill raker numbers in these species by comparing the copy numbers (presence or absence), syntenies and genomic structures of candidate genes (genes known to control the development of gill rakers) between fish with different numbers of gill rakers. Several references reported similar methods of searching for known functional genes in other genomes. For example, Liu et al. (2016) checked genes known to cause scale loss when mutated (*eda*, *edar*, *fgfr*, *lef1* and *tcf7*) in channel catfish and found that these genes were all present and expressed in channel catfish, although this species is scaleless. The authors further conducted comparative transcriptome analysis between scaled and scaleless fishes. Yang et al. (2016) reported that “A previous study has indicated that mutations in the ectodysplasin-A receptor (*Edar*) encoding locus can lead to complete scale loss in fish such as medaka. For this reason, two copies of *Edar* gene (named as *Edar1* and *Edar2*, respectively) in the three *Sinocyclocheilus* genomes were identified and checked.”

References:

- (1) Sanderson, S. L., Cheer, A. Y., Goodrich, J. S., Graziano, J. D. & Callan, W. T. Crossflow filtration in suspension-feeding fishes. *Nature* **412**, 439-441 (2001).
- (2) Magnuson, J. J. & Heitz, J. G. Gill raker apparatus and food selectivity among mackerels, tunas, and dolphins. *Fish Bull. Natl. Oc. At.* **69**, 361-370 (1971).
- (3) Roesch, C., Lundsgaard-Hansen, B., Vonlanthen, P., Taverna, A. & Seehausen, O. Experimental evidence for trait utility of gill raker number in adaptive radiation of a north temperate fish. *J. Evol. Biol.* **26**, 1578-1587 (2013).
- (4) Abumandour, M. & El-Bakary, N. E. Gill Morphology in Two Bottom Feeder Mediterranean Sea Fishes: Grey Gurnard Fish (*Eutrigla gurnardus*, Linnaeus, 1758) and Striped Red Mullet Fish (*Mullus barbatus surmuletus*, Linnaeus, 1758) by Scanning Electron Microscopy. *International Journal of Morphology* **35** (2017).
- (5) Harris, M. P. et al. Zebrafish *eda* and *edar* mutants reveal conserved and ancestral roles of ectodysplasin signaling in vertebrates. *PLoS Genet.* **4**, e1000206 (2008).
- (6) Glazer, A. M., Cleves, P. A., Erickson, P. A., Lam, A. Y. & Miller, C. T. Parallel developmental genetic features underlie stickleback gill raker evolution. *EvoDevo* **5**, 19 (2014).

- (7) Liu, Z. et al. The channel catfish genome sequence provides insights into the evolution of scale formation in teleosts. *Nat. Commun.* 7, 11757 (2016).
- (8) Yang, J. et al. The *Sinocyclocheilus* cavefish genome provides insights into cave adaptation. *BMC Biol.* 14(1), 1 (2016).

** See Nature Research's author and referees' website at www.nature.com/authors for information about policies, services and author benefits

COMMSBIO - This email has been sent through the Springer Nature Tracking System NY-610A-NPG&MTS

Confidentiality Statement:

This e-mail is confidential and subject to copyright. Any unauthorised use or disclosure of its contents is prohibited. If you have received this email in error please notify our Manuscript Tracking System Helpdesk team at <http://platformsupport.nature.com> .

Details of the confidentiality and pre-publicity policy may be found here <http://www.nature.com/authors/policies/confidentiality.html>

Privacy Policy | Update Profile

DISCLAIMER: This e-mail is confidential and should not be used by anyone who is not the original intended recipient. If you have received this e-mail in error please inform the sender and delete it from your mailbox or any other storage mechanism. Springer Nature America, Inc. does not accept liability for any statements made which are clearly the sender's own and not expressly made on behalf of Springer Nature America, Inc. or one of their agents.

Please note that neither Springer Nature America, Inc. or any of its agents accept any responsibility for viruses that may be contained in this e-mail or its attachments and it is your responsibility to scan the e-mail and attachments (if any).

REVIEWERS' COMMENTS:

Reviewer #1 (Remarks to the Author):

The manuscript was improved with an extra editing service, and the authors answered and resolved most of the mentioned issues. However, the authors didn't pay much attention to the problems in Figure 1, which should be simplified since most of the phylogenetic tree has been published in many references (such as a recent extensive report in Hughes et al., 2018, PNAS, 115:6249-6254). Meanwhile, the numbers in the figure are too small. The authors are recommended to enlarge them for a better vision.

Reviewer #2 (Remarks to the Author):

I went through the revised paper carefully. My major concern regards the de novo gene prediction and the function of edar and the other results interpretation for this paper that raises some questions most of which are addressed by the authors. The paper is closely and clearly written, and I have no other questions.

REVIEWERS' COMMENTS:

Reviewer #1 (Remarks to the Author):

The manuscript was improved with an extra editing service, and the authors answered and resolved most of the mentioned issues. However, the authors didn't pay much attention to the problems in Figure 1, which should be simplified since most of the phylogenetic tree has been published in many references (such as a recent extensive report in Hughes et al., 2018, PNAS, 115:6249-6254). Meanwhile, the numbers in the figure are too small. The authors are recommended to enlarge them for a better vision.

R:

We have now followed the reviewers recommendations to make the figure more readable. We removed further species from Fig. 1 (now containing only 15 species) and changed the coloring and font size. To show the big picture to the readers we would like to put the trees from the first submission and first revision (121 and 34 species) in the supplement.

Regarding the phylogenomic approach we do not agree with the reviewer that we do not provide new insights. In our opinion our trees are not just a repetition of known findings as we used whole genome alignments and filtered for non-coding sequence. Most studies in large teleost phylogenomic projects still use protein coding sequence to infer phylogenetic trees, such as the study the reviewer cites (If you look at Fig. 2 of Hughes et al., they used 555,288 nt characters translating into 185,096 aa characters). Our tree corroborates most results of Hughes et al. with a different data type (non-coding sequence) and more data (several million aligned nt).

More importantly and in contrast to Hughes et al. 2018, our tree showed high statistical support for nearly all nodes of the hard to resolve Percomorphacae. If you look at the support values of the Percomorphacae clade in Fig. 2 of Hughes et al. you will find that they have many nodes that are not supported to 100% and many with values even below 70%, while in our trees we obtained the highest possible support values for all nodes of the Percomorphacae. Although

there are many differences in number of taxons sampled and software used between our study and Hughes et al. 2018, this may imply that non-coding sequence is the better choice for inferring phylogenomic species trees in teleosts, like it has been shown in bird phylogenomics (Jarvis et al. 2014).

Reviewer #2 (Remarks to the Author):

I went through the revised paper carefully. My major concern regards the de novo gene prediction and the function of edar and the other results interpretation for this paper that raises some questions most of which are addressed by the authors. The paper is closely and clearly written, and I have no other questions.

R:

We would like to thank both reviewers for their valuable comments.